# Circulating Tumor Cells and TWIST Expression in Patients with Metastatic Gastric Cancer: A Preliminary Study

**DOI:** 10.3390/jcm10194481

**Published:** 2021-09-29

**Authors:** Joon Hyung Jhi, Gwang Ha Kim, Su Jin Park, Dong Uk Kim, Moon Won Lee, Bong Eun Lee, Chae Hwa Kwon, Yoon-Kyoung Cho

**Affiliations:** 1Department of Internal Medicine, Pusan National University College of Medicine, Busan 49241, Korea; jhjhimd@naver.com (J.H.J.); amlm3@hanmail.net (D.U.K.); neofaceoff@hanmail.net (M.W.L.); bongsul@hanmail.net (B.E.L.); 2Biomedical Research Institute, Pusan National University Hospital, Busan 49241, Korea; soo-jean@hanmail.net (S.J.P.); chkwon@pusan.ac.kr (C.H.K.); 3Department of Biomedical Engineering, Ulsan National Institute of Science and Technology (UNIST), Ulsan 44919, Korea; ykcho@unist.ac.kr; 4Center for Soft and Living Matter, Institute for Basic Science (IBS), Ulsan 44919, Korea

**Keywords:** circulating tumor cells, stomach neoplasms, TWIST, prognosis

## Abstract

Background and Aims: The clinical significance of circulating tumor cells (CTCs) and TWIST expression in CTCs remains unelucidated in patients with gastric cancer (GC). Here, we evaluated CTCs and TWIST expression in CTCs and explored their correlation with prognosis in patients with metastatic GC. Methods: Peripheral blood samples were prospectively obtained from 31 patients with metastatic GC between September 2017 and December 2018, prior to treatment. CTCs were detected using a centrifugal microfluidic system and CTCs positive for TWIST immunostaining were defined as TWIST (+) CTCs. Results: CTCs and TWIST (+) CTCs were detected in 25 (80.6%) and 24 (77.4%) of the 31 patients, respectively. CTC count in patients with first diagnosis of metastatic cancer tended to be higher than that in those with recurrent metastatic cancer, but TWIST (+) CTC count was not different between the two groups. There was no difference in CTC and TWIST (+) CTC counts according to histopathologic type, peritoneal dissemination, hematogenous metastasis, serum tumor makers, or response to first-line chemotherapy. Patients with CTCs > 7.5/7.5 mL of blood showed shorter overall survival (OS) than those with CTCs ≤ 7.5/7.5 mL of blood (*p* = 0.049). Additionally, patients with TWIST (+) CTCs > 2.5/7.5 mL of blood tended to show shorter OS than those with TWIST (+) CTCs ≤ 2.5/7.5 mL of blood (*p* = 0.105). Conclusions: Our study demonstrated that high levels of CTCs and TWIST (+) CTCs were associated with worse OS.

## 1. Introduction

Gastric cancer (GC) is the fifth most-common malignant tumor and the third-leading cause of cancer-related mortality worldwide [1]. It is estimated that 1,033,700 new cases of GC and 782,700 GC-related deaths occurred worldwide in 2018, which accounted for 6.1% of the total new cancer cases and 8.2% of the total cancer-related deaths [1]. Although advances in diagnosis, surgery, and chemotherapeutics have improved the clinical outcome of GC patients, the 5-year survival rate remains <30%. Distant metastasis and recurrence are the leading causes of GC-related mortality [2].

Circulating tumor cells (CTCs), defined as tumor cells that originate from either primary or metastasis sites and circulate freely in the peripheral blood of patients with cancer [3], are reported to serve as a biomarker for early detection of cancer and provide useful information about treatment response and prognosis following surgery or chemotherapy in GC patients [4,5,6]. Various methods of CTC detection have been reported based on molecular and biological approaches, including flow cytometry and reverse transcriptase-polymerase chain reaction (RT-PCR), in GC patients [7,8]. Recently, a centrifugal microfluidic system using a fluid-assisted separation technique (FAST) was developed for CTC detection [9,10], and we previously suggested the potential role of FAST-based CTC detection as an early diagnostic biomarker in GC [11].

Epithelial–mesenchymal transition (EMT) plays a key role in tumor progression, metastasis, and recurrence [12]. EMT enables epithelial cells to gain enhanced invasive potential by losing their epithelial features and acquiring a mesenchymal phenotype; in other words, in the EMT process of tumor cells, the expression of epithelial markers (e.g., epithelial cell adhesion molecule [EpCAM] and cytokeratins [CKs]) is downregulated, while that of mesenchymal markers (e.g., vimentin and TWIST) is upregulated [12]. Among the mesenchymal makers, TWIST is known to be a development gene that plays a key role in the induction of EMT. Loss of TWIST expression hinders the intravasation of metastatic tumor cells into the blood circulation [13]. Although some studies have suggested that TWIST is an independent prognostic marker [14,15,16], studies on the role of TWIST in patents with GC are limited.

CTCs are correlated with shorter overall survival in patients with metastatic disease [6,17,18]. However, there have been few reports on the clinical significance of CTCs and TWIST expression in CTCs in patients with metastatic GC. We hypothesized that CTCs, especially TWIST-expressing CTCs, are associated with prognosis in patients with metastatic GC. Therefore, we evaluated CTCs and TWIST expression in CTCs and explored their correlation with prognosis in patients with metastatic GC.

## 2. Methods

### 2.1. Study Design (Patients and Sample)

Patients with metastatic GC who had not been diagnosed with other past or current malignancies at the Pusan National University Hospital (Busan, Korea) were analyzed using prospectively collected data. The study protocol was reviewed and approved by the Institutional Review Board of the Pusan National University Hospital (H-1412-011-024), and written informed consent was obtained from all the patients before they underwent blood sampling. The study was performed in accordance with the Declaration of Helsinki. In total, 32 consecutive patients with metastatic GC were enrolled between September 2017 and December 2018. One patient was excluded from the analysis because the blood sample was obtained during chemotherapy. The patients were divided into two groups; those who were first diagnosed with GC with distant metastasis (the de novo group; *n* = 14) and those in whom recurrent distant metastasis occurred after gastrectomy (the recurrence group; *n* = 17) (Figure 1).

The evaluation included collection of medical history, physical examination, routine blood tests, analysis of serum tumor markers (carcinoembryonic antigen [CEA] and cancer antigen 19-9 [CA19-9]), and imaging studies using chest and abdominal computed tomography (CT) and/or positron-emission tomography with CT for tumor staging. Staging was assessed according to the American Joint Committee on Cancer TNM staging for GC (the eighth edition) [19]. Peripheral blood was collected prior to the start of chemotherapy. Progression of the disease following chemotherapy was determined according to the response evaluation criteria in solid tumors (RECIST 1.1 criteria) [20]. Overall survival was defined as the time from the date of registration to the date of death from any cause. Patient data were collected until 29 February 2020. The biospecimens and data used for this study were provided by the Biobank of Pusan National University Hospital, a member of the Biobank Network.

### 2.2. Isolation and Enumeration of CTCs

CTCs were isolated according to the protocol specified in our previous reports [11,21]. Peripheral blood samples were collected prior to chemotherapy. The initial 2 mL of blood drawn was discarded to prevent contamination with skin epithelial and mesenchymal cells and then 3–5 mL of peripheral blood was collected in K2-EDTA tubes and inverted 10 times immediately. The blood samples were tested within 8 h to minimize cell damages. Then, FAST was performed to isolate CTCs from whole blood without pretreatment such as dilution or dissolution of red blood cell lysis. Before CTC isolation, the disc surface was passivated with 1% bovine serum albumin and then washed with phosphate-buffered saline (PBS). After the surface passivation, 3 mL whole blood was introduced to the disc, and CTCs were isolated on a track-etched polycarbonate membrane using a spinning disc device.

To identify the CTCs isolated from the blood sample, the disc was immunostained [11,21]. First, the captured cells were fixed with 4% formaldehyde for 20 min at room temperature. The fixed cells were then permeabilized with 0.1% Triton X-100 in PBS for 5 min and subsequently washed with PBS. Next, the samples were blocked with 20 μg/mL immunoglobulin G (polyclonal human IgG, R&D system, Minneapolis, MN, USA) for 20 min, followed by staining with the following antibodies. An anti-CD45 PE-Alexa Fluor (H130; Life Technologies, Carlsbad, CA, USA) was applied for 20 min to stain the white blood cells, and samples were then washed with 0.01% Tween 20 in PBS. For staining the CTCs, a mixture of fluorescein isothiocyanate-conjugated anti-cytokeratin (CAM5.2; BD. Franklin Lakes, NJ, USA), Alexa Fluor 488-conjugated anti-panCK (AE1/AE3; eBioscience, San Diego, CA, USA), fluorescein isothiocyanate-conjugated anti-EpCAM (9C4; BioLegend, San Diego, CA, USA), and TWIST (Twist2Cla, BMR, Chiba, Japan) was added to the membrane and incubated for 20 min. After washing with 0.01% Tween 20 in PBS, Alexa Fluor 647-conjugated secondary antibody (anti-Goat, Invitrogen, Carlsbad, CA, USA) was added for TWIST detection, incubated for 20 min, and washed. Finally, the membrane was mounted on a glass slide with mounting medium containing fluorescent nucleic acid dye 4,6-diamidino-2-phenylindole (DAPI). To visualize CTCs on the membrane, slides were scanned using the Eclipse Ti-E fluorescent microscope (Nikon, Tokyo, Japan) at 40× magnification. The captured cells were identified as CTCs if they were CK+ or EpCAM+, CD45− and DAPI+, and their diameter was >8 μm [11,21]. CTCs positive for TWIST immunostaining were defined as TWIST (+) CTCs (Figure 2). Results were quantified as the number of CTCs/7.5 mL of whole blood.

### 2.3. Statistical Analysis

Data are expressed as median and ranges. Differences in the total CTC and TWIST (+) CTC counts according to the clinicopathologic features were evaluated using the Wilcoxon rank-sum test. The association between CTCs or TWIST (+) CTCs and various clinicopathologic characteristics was analyzed using Fisher exact test or x^2^ test. The optimal cut-off values for CTCs and TWIST (+) CTCs were determined using the maximally selected rank statistics. The Kaplan–Meier method was used for survival analysis according to the level of CTCs and TWIST (+) CTCs, and the differences in survival were examined using the log-rank test. Cox proportional hazards models were used for multivariate analyses, and the hazard ratio (HR) was estimated. The statistical analyses were conducted using the language R (http://cran.r-project.org) version 4.0.0. A *p* < 0.05 was considered statistically significant.

## 3. Results

### 3.1. Baseline Clinicopathologic Characteristics of Patients with Metastatic GC

Clinicopathologic characteristics of the 31 patients with metastatic GC are summarized in Table 1. The patients comprised 22 men and 9 women, with a median age of 63 years (range, 42–87 years). Twelve patients had hypertension; 2 had diabetes mellitus; 2 had chronic hepatitis B; 2 had a previous history of pulmonary tuberculosis; 1 had Parkinson’s disease; and 1 had benign prostate hypertrophy. Fourteen patients were in an inoperable state due to metastasis to other organs or peritoneum at the first diagnosis (de novo group), and 17 patients had recurrence in the other organs or peritoneum following gastrectomy (recurrence group). The median duration from surgical resection to recurrence, in the recurrence group, was 11.6 months (range, 2–37 months). Four patients remained alive at the time of this analysis. Histopathologically, 13 tumors were intestinal type, and 18 were diffuse type. Peritoneal dissemination was present in 19 patients and hematogenous metastasis in 17 patients. Twenty-six patients received systemic chemotherapy; the chemotherapy regimens included 5-fluorouracil (5-FU) or capecitabine + cisplatin +/− trastuzumab, 5-FU + irinotecan, 5-FU + oxaliplatin, or paclitaxel + cisplatin. Based on the response evaluation criteria in solid tumors (RECIST 1.1 criteria) [20], 4 patients achieved partial remission, 9 had stable disease, and 13 had progressive disease after first-line chemotherapy.

### 3.2. Incidence of CTCs in Patients with Metastatic GC

CTC and TWIST (+) CTC counts of each patient with GC are shown in Figure 3. CTCs and TWIST (+) CTCs were identified in 25 (80.6%) and 24 (77.4%) of the 31 patients, respectively; in 11 (78.6%) and 11 (78.6%) patients from the de novo group, respectively, and in 14 (82.3%) and 13 (76.5%) patients from the recurrence group, respectively. The median CTC and TWIST (+) CTC counts were 5.0 (range, 0–62.4) and 2.5 (range, 0–62.4)/7.5 mL of blood, respectively, in all patients; 20.0 (range, 0–47.5) and 5.0 (0–40.0)/7.5 mL of blood in the de novo group, respectively; and 2.5 (range, 0–62.4) and 2.5 (0–62.4)/7.5 mL of blood in the recurrence group, respectively.

### 3.3. Association between CTCs and Clinicopathologic Characteristics

Table 2 summarizes the CTC and TWIST (+) CTC counts according to the clinicopathologic characteristics of the 31 patients with metastatic GC. The CTC count in the de novo group was higher than that in the recurrence group, although the difference was not statistically significant (20.0 [0–47.5] vs. 2.5 [0–62.4], *p* = 0.113). There were no differences in the CTC and TWIST (+) CTC counts according to age, sex, histopathologic type, peritoneal dissemination, hematogenous metastasis, serum tumor makers, or response to first-line chemotherapy.

To investigate whether the number of CTCs and TWIST (+) CTCs was associated with the clinicopathologic characteristics of the patients, we analyzed the correlation based on the cut-off values of CTC and TWIST (+) CTC counts. Thresholds of >7.5 and >2.5/7.5 mL of blood for CTCs and TWIST (+) CTCs, respectively, was selected as the cut-off values in our correlation analysis (Appendix A). The clinicopathologic characteristics of patients, according to these cut-off values of CTCs and TWIST (+) CTCs, are presented in Table 3. CTCs at >7.5/7.5 mL of blood was observed more frequently in the de novo group and in patients with increased serum CA19-9, but the differences did not reach statistical significance (*p* = 0.055 and *p* = 0.071, respectively). TWIST (+) CTCs at >2.5/7.5 mL of blood was also observed more frequently in the de novo group, which also did not reach statistical significance (*p* = 0.119). There were no significant differences in the other clinicopathologic characteristics based on the selected cut-off levels of CTCs and TWIST (+) CTCs.

### 3.4. Survival Outcomes According to the Level of CTCs

All patients were categorized into high-CTC (>7.5/7.5 mL of blood) and low-CTC (≤7.5/7.5 mL of blood) groups during the follow-up period. To investigate the relationship between CTCs and clinical outcomes, we analyzed the overall survival according to the level of CTCs. The median follow-up period was 12 months (range, 1–30 months), and there were 27 (87.1%) cases of death. The overall survival in the high-CTC group was significantly shorter than that in the low-CTC group (*p* = 0.049; Figure 4). When the patients were categorized into high TWIST (+) CTC (>2.5/7.5 mL of blood) and low TWIST (+) CTC (≤2.5/7.5 mL of blood) groups, the overall survival in the high TWIST (+) CTC group was shorter than that in the low TWIST (+) CTC group; however, the difference did not reach statistical significance (*p* = 0.105).

### 3.5. Analysis of Other Prognostic Factors

Uni- and multivariate analyses were performed to evaluate the effect of clinicopathologic factors on overall survival. Older age (>65 years), de novo status, and elevated CA 19-9 (>37 U/mL) were correlated with shorter overall survival. Multivariate analyses showed that peritoneal dissemination, elevated CA 19-9 (>37 U/mL), and the non-administration of chemotherapy were independent prognostic factors (Table 4).

## 4. Discussion

In the present study, we evaluated CTCs and TWIST expression in CTCs using FAST in 31 patients with metastatic GC; CTCs and TWIST (+) CTCs were detected in 25 (80.6%) and 24 (77.4%) patients, respectively. Patients with CTCs > 7.5/7.5 mL of blood showed shorter overall survival than those with CTCs ≤ 7.5/7.5 mL of blood. In addition, patients with TWIST (+) CTCs > 2.5/7.5 mL of blood tended to show shorter overall survival than those with TWIST (+) CTCs ≤ 2.5/7.5 mL of blood. To the best of our knowledge, this is the first study to detect TWIST (+) CTCs using FAST and to evaluate their correlation with clinicopathologic characteristics and overall survival in patients with metastatic GC.

Recently, CTCs have been suggested to be promising biomarkers for early diagnosis and predictor of survival and prognosis in GC patients [3,11]. In our previous study, we measured CTCs using FAST in 116 GC patients and 31 healthy individuals [11]. When CTCs ≥ 2/7.5 mL of blood was set as the cut-off value, the sensitivity and specificity of differentiating GC patients from healthy controls were 85.3% and 90.3%, respectively. Although the CTCs were not associated with any clinicopathologic feature including histopathologic type, mucin phenotype, or staging, we suggested the potential of CTCs as an early diagnostic biomarker for GC. In another recent study, using a wedge-shaped microfluidic chip, CTCs were associated with tumor differentiation, lymphovascular invasion, and staging [22]. In the present study, we only included patients with metastatic GC; there was no difference in CTC counts according to histopathologic type, peritoneal dissemination, and hematogenous metastasis. These differences in the association between CTCs and clinicopathologic parameters may be due to heterogeneity in the baseline clinicopathologic background.

In a prospective study that evaluated CTCs in 251 patients with advanced GC using the CellSearch^TM^ system, CTCs were detected in 62 patients (62/103, 60.2%) in the non-resectable group, and the overall survival rate was significantly lower in patients with CTCs than in those without CTCs in the non-resectable group [6]. In a meta-analysis that included 26 studies comprising 2566 GC patients, the frequency of CTC detection was significantly related to the disease-free and overall survival of patients [23]. Similarly, in the present study, CTCs were detected in 80.6% of patients with metastatic GC, and a high CTC number (>7.5/7.5 mL of blood) was associated with poor overall survival. Our results are also consistent with the results of a previous study in which the overall survival rate of patients with CTCs > 5/7.5 mL of blood tended to be lower than that of the patients with CTCs ≤ 5/7.5 mL of blood [24].

CellSearch^TM^ system is the most commonly used and the only United States Food and Drug Administration-approved CTC detection test applicable in patients with various cancers. This test system requires the enumeration of epithelial cells, which are separated from the blood using EpCAM antibody-coated magnetic beads and identified with fluorescently labeled antibodies against CKs and a fluorescent nuclear stain. However, EpCAM-based enrichment cannot detect CTC subpopulations that have already or partially undergone EMT [25,26]. In contrast, FAST is a size-based isolation kit, with several advantages compared to the other CTC detection methods, including a user-friendly, cost-effective, and efficient CTC capture technique [9,10]. Furthermore, FAST enables further downstream molecular analysis for the detection of TWIST (+) CTCs using immunostaining. Therefore, we could investigate TWIST expression in the CTCs using FAST in patients with metastatic GC. EMT is a multi-step process that plays a key role in metastasis, cancer progression, and therapeutic resistance. Therefore, we predicted that TWIST expression in CTCs would be strongly associated with poor prognosis and response to chemotherapy in patients with metastatic GC. In the present study, although the difference was not statistically significant, TWIST (+) CTCs tended to be associated with shorter survival. This result is similar to the results of previous studies that patients with TWIST overexpression in CTCs had lower overall survival in breast cancer and hepatocellular carcinoma [27,28]. However, TWIST (+) CTCs were not associated with poor response to chemotherapy. We believe that these results may be caused by the relatively small sample size of our study and the use of various chemotherapy regimens, and that examination of larger cohorts may reveal a more significant impact of TWIST (+) CTCs on these clinical parameters.

CTCs can also be used to monitor response to chemotherapy [4]. The measurement of CTCs using peripheral blood is non-invasive and can be performed repeatedly, and the periodic monitoring of CTCs may be useful to predict the efficacy of chemotherapy. Although CEA and CA19-9 are frequently used as markers in GC patients, these markers are elevated in less than 40% of patients with advanced GC, and they temporary increase following chemotherapy; therefore, the sensitivity and specificity of assays to detect these markers are insufficient for prognosis [29]. In the present study, CEA and CA19-9 were elevated in 13 (41.9%) and 18 (58.0%) patients with metastatic GC, respectively. In a study that measured CTC levels serially in 52 patients with advanced GC, patients with ≥4 CTCs/7.5 mL of blood at 2- and 4-weeks post-chemotherapy showed shorter median progression-free survival and overall survival [30]. In another study CTC levels were measured in 136 patients with advanced GC, and patients with ≥3 CTCs/7.5 mL of blood after chemotherapy had shorter progression free survival and overall survival, and elevated levels of CTCs after chemotherapy was associated with ineffective therapeutic response [26]. We also found that CTC and TWIST (+) CTC counts decreased following chemotherapy compared to before chemotherapy in some patients (data not shown). We intend to evaluate the potential of CTCs and TWIST (+) CTCs as useful supplementary markers to monitor the response to chemotherapy in GC patients in the future.

Our study has several limitations that need to be addressed. First, the number of patients included in the study was relatively small and the composition of patients was somewhat heterogeneous. We plan to include more patients with non-metastatic GC as well as metastatic GC and then to analyze the roles of CTCs and TWIST (+) CTCs in predicting long-term outcomes. Second, other mesenchymal markers such as vimentin or stem cell markers (such as CD44) were not studied along with TWIST in the CTCs. Third, we detected CTCs based on an EpCAM-based enrichment technique and evaluated TWIST expression in these CTCs. Based on the expression of epithelial and mesenchymal markers, CTCs is classified into three subpopulations: epithelial CTCs, biphenotypic epithelial/mesenchymal CTCs, and mesenchymal CTCs [31]. We did not evaluate mesenchymal CTCs without EpCAM expression, which are reported to be associated with metastasis and disease progression [31]. Finally, CTCs detected in patients with metastatic GC are unlikely to be organ-specific. Although we excluded other current malignancies through the examinations mentioned earlier, it is possible that undetected malignancies may be present in other organs and that these could be a source of the CTCs.

In conclusion, CTCs and TWIST (+) CTCs were detected in 4/5 of patients with metastatic GC, and high levels of CTCs and TWIST (+) CTCs was associated with worse overall survival. Our study provides promising results for the use of CTCs and TWIST (+) CTCs as prognostic biomarkers in patients with metastatic GC. The additional role of CTCs and TWIST (+) CTCs as biomarkers for the prediction of response to chemotherapy should be investigated in large, prospective, long-term follow-up studies.

## Figures and Tables

**Figure 1 jcm-10-04481-f001:**
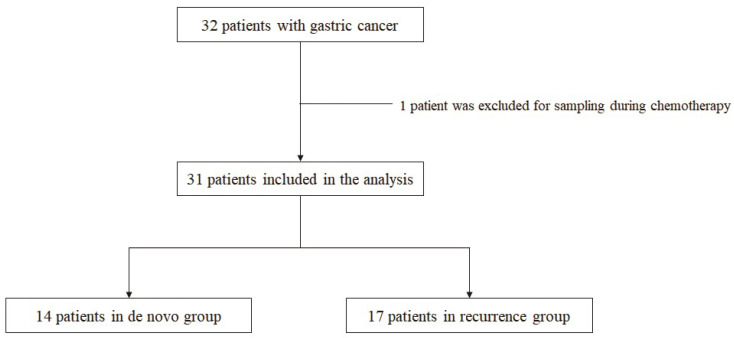
Flowchart of the patients included in the study.

**Figure 2 jcm-10-04481-f002:**
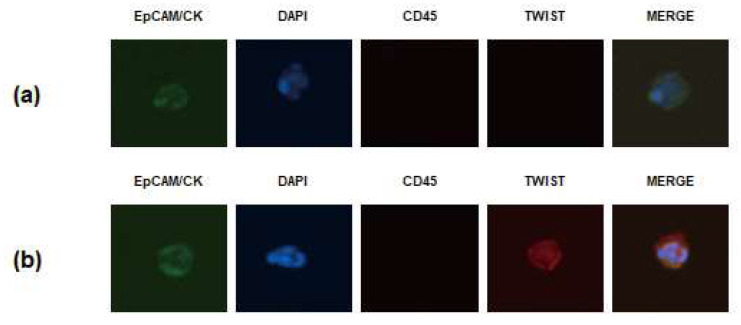
Circulating tumor cells (CTCs) detected in patients with metastatic gastric cancer. (**a**) Representative images of CTCs negative for TWIST immunostaining. (**b**) Representative images of CTCs positive for TWIST immunostaining.

**Figure 3 jcm-10-04481-f003:**
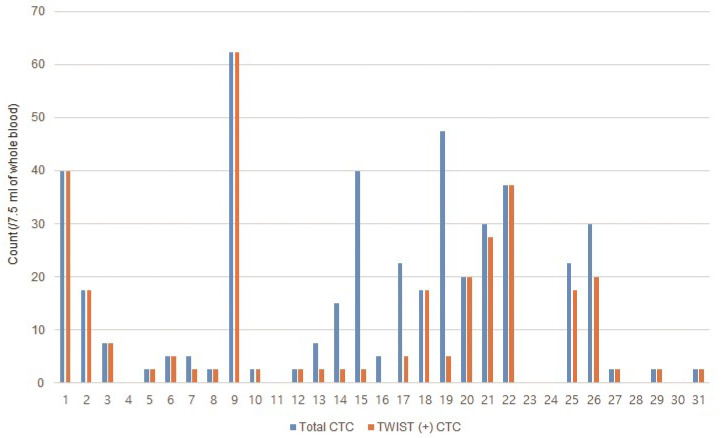
Circulating tumor cell (CTC) and TWIST (+) CTC counts in the 31 patients with metastatic gastric cancer.

**Figure 4 jcm-10-04481-f004:**
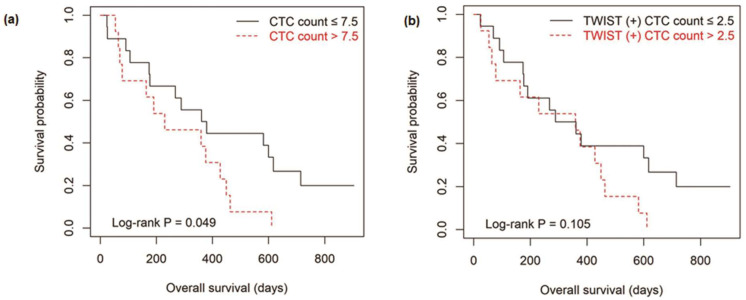
(**a**) Overall survival curves of 18 patients with ≤7.5 circulating tumor cells (CTCs) and 13 patients with >7.5 CTCs/7.5 mL of whole blood. (**b**) Overall survival curves of 18 patients with ≤2.5 TWIST (+) CTCs and 13 patients with >2.5 TWIST (+) CTCs/7.5 mL of whole blood.

**Table 1 jcm-10-04481-t001:** Clinicopathologic characteristics of the 31 patients with metastatic gastric cancer.

Median age, years (range)	63 (42–87)
Sex	
Male	22
Female	9
Disease status	
De novo	14
Recurrence	17
Histopathologic type	
Intestinal type	13
Diffuse type	18
Peritoneal dissemination	
Absent	12
Present	19
Hematogenous metastasis	
Absent	14
Present	17
Serum CEA	
≤5 ng/mL	18
>5 ng/mL	13
Serum CA19-9	
≤37 U/mL	13
>37 U/mL	18
Chemotherapy	
Not administered	5
Administered	26

**Table 2 jcm-10-04481-t002:** Circulating tumor cell (CTC) and TWIST (+) CTC counts according to the clinicopathologic characteristics of 31 patients with metastatic gastric cancer.

	CTC Count (/7.5 mL of Whole Blood)	*p* Value *	TWIST (+) CTC Count (/7.5 mL of Whole Blood)	*p* Value *
Age		0.627		0.398
≤65 years	11.3 (0–62.4)		3.8 (0–62.4)	
>65 years	5.0 (0–47.5)		2.5 (0–20.0)	
Sex		0.965		0.946
Male	6.3 (0–40.0)		2.5 (0–40.0)	
Female	5.0 (0–62.4)		2.5 (0–62.4)	
Disease status		0.113		0.368
De novo	20.0 (0–47.5)		5.0 (0–40.0)	
Recurrence	2.5 (0–62.4)		2.5 (0–62.4)	
Histopathologic type		0.284		0.216
Intestinal type	2.5 (0–40.0)		2.5 (0–20.0)	
Diffuse type	11.3 (0–62.4)		3.8 (0–62.4)	
Peritoneal dissemination		0.566		0.337
Absent	7.5 (0–47.5)		3.8 (9–40.0)	
Present	5.0 (0–62.4)		2.5 (0–62.4)	
Hematogenous metastasis		0.703		0.790
Absent	5.0 (0–62.4)		2.5 (0–62.4)	
Present	7.5 (0–47.5)		2.5 (0–40.0)	
Serum CEA		0.558		0.564
≤5 ng/mL	6.3 (0–62.4)		3.8 (0–62.4)	
>5 ng/mL	2.5 (0–40.0)		2.5 (0–37.3)	
Serum CA19-9		0.175		0.155
≤37 U/mL	2.5 (0–47.5)		2.5 (0–27.5)	
>37 U/mL	16.3 (0–62.4)		3.8 (0–62.4)	
Response to chemotherapy ^†^		0.717		0.832
Partial remission/stable disease	5.0 (0–62.4)		2.5 (0–62.4)	
Progressive disease	5.0 (0–40.0)		2.5 (0–40.0)	

Data are expressed by median (range). * *p* values are from Wilcoxon rank-sum test. ^†^ Twenty-six patients underwent chemotherapy.

**Table 3 jcm-10-04481-t003:** Clinicopathologic characteristics of the 31 patients with metastatic gastric cancer according to the level of circulating tumor cells (CTC) and TWIST (+) CTCs.

	CTC Count (/7.5 mL of Whole Blood)		*p* Value *	TWIST (+) CTC Count (/7.5 mL of Whole Blood)		*p* Value *
	≤7.5 (*n* = 18)	>7.5 (*n* = 13)		≤2.5 (*n* = 18)	>2.5 (*n* = 13)	
Age			0.284			0.284
≤65 years	9	9		9	9	
>65 years	9	4		9	4	
Sex			1.000			1.000
Male	13	9		13	9	
Female	5	4		5	4	
Disease status			0.055			0.119
De novo	5	9		6	8	
Recurrence	13	4		12	5	
Histopathologic type			0.284			0.284
Intestinal type	9	4		9	4	
Diffuse type	9	9		9	9	
Peritoneal dissemination			0.981			0.470
Absent	7	5		6	6	
Present	11	8		12	7	
Hematogenous metastasis			0.524			0.524
Absent	9	5		9	5	
Present	9	8		9	8	
Serum CEA			0.739			0.284
≤5 ng/mL	10	8		9	9	
>5 ng/mL	8	5		9	4	
Serum CA19-9			0.071			0.284
≤37 U/mL	10	3		9	4	
>37 U/mL	8	10		9	9	
Response to chemotherapy ^†^			0.420			1.000
Partial remission/stable disease	9	4		8	5	
Progressive disease	7	6		8	5	

* *p* values are from Fisher exact test or x^2^ test. ^†^ Twenty-six patients underwent chemotherapy.

**Table 4 jcm-10-04481-t004:** Univariate and multivariate analyses of clinicopathologic factors for overall survival in the 31 patients with metastatic gastric cancer.

Clinicopathologic Factors	Univariate Analysis	Multivariate Analysis	
	*p* Value	HR (95% CI)	*p* Value
Sex (M vs. F)	0.956	0.860 (0.279−2.651)	0.793
Age (≤65 years vs. >65 years)	0.033	0.428 (0.127−1.436)	0.169
Disease status (de novo vs. recurrence)	0.008	2.458 (0.561−10.777)	0.233
Histopathologic type (intestinal vs. diffuse)	0.051	2.756 (0.976−7.781)	0.056
Peritoneal dissemination	0.331	4.293 (1.155−15.962)	0.030
Hematogenous metastasis	0.183	1.281 (0.440−3.732)	0.650
Serum CEA (≤5 ng/mL vs. >5 ng/mL)	0.711	0.997 (0.328−3.029)	0.996
Serum CA19-9 (≤37 U/mL vs. >37 U/mL)	0.005	7.531 (2.301−24.645)	0.001
Chemotherapy	0.144	0.154 (0.028−0.853)	0.032
CTC count/7.5 mL of whole blood (≤7.5 vs. >7.5)	0.054	0.848 (0.222−3.245)	0.810
TWIST (+) CTC count/7.5 mL of whole blood (≤2.5 vs. >2.5)	0.111	0.723 (0.167−3.127)	0.664

HR, hazard ratio; CI, confidence interval.

## Data Availability

The data presented in this study are available on request from the corresponding author.

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
