# Peer review of "Circulating Tumor Cells and TWIST Expression in Patients with Metastatic Gastric Cancer: A Preliminary Study"

_jcm, 2021, doi:10.3390/jcm10194481_

Round 1

Reviewer 1 Report

Following the remarks of the first submission the authors have:

- modulated their statements by modifying the title, notably: “preliminary study” was added to the title

- performed correlation between clinicopathological characteristics detailed in table 1 and OS with univariate and multivariate analyses. The latter show that in univariate analysis the total number of CTCs is at the limit of significance concerning OS whereas the serum level of CA19-9 is very significant (p=0.005). On the other hand, in multivariate analysis, the number of CTCs was no longer associated with overall survival, whereas the serum CA19-9 level remained highly significant.

Nevertheless, the limitations related to the low number of patients and their heterogeneity, although described in the text, remain.

Author Response

Response to Reviewer #1’s Comments

We agree with the Reviewer 1’s comment. However, we cannot include more GC patients and collect their follow-up data at this point. Therefore, we have added the follow sentence in the Discussion section. We sincerely request the Reviewer 1’s consideration.

Our study has several limitations that need to be addressed. First, the number of patients included in the study was relatively small and the composition of patients was somewhat heterogeneous. We plan to include more patients with non-metastatic GC as well as metastatic GC and then to analyze the roles of CTCs and TWIST (+) CTCs in predicting long-term outcomes.

Reviewer 2 Report

In the manuscript entitled - “Circulating Tumor Cells and TWIST Expression in Patients with Metastatic Gastric Cancer: A Preliminary Study”, the authors have addressed a relevant topic on utilizing TWIST+ CTCs as prognostic markers for metastatic gastric cancers. While there are already studies on TWIST+ CTCs, the subject selection from a different geographical location makes this study interesting. The authors have now included a detailed discussion on the limitations of the study. The study has been performed comprehensively on a small cohort of subjects and should be published as a preliminary study. However, there are few concerns that need to be clarified and addressed in detail before publication:

  1. The authors should also include other studies in the discussion that describe CTCs and TWIST+ CTCs as prognostic biomarkers. Moreover, the authors should mention at the start and in relevant places the uniqueness of cohort from a different geographical location as the strength of their study.
  2. The authors should also report if the 31 enrolled patients have any underlying disease or co-morbidity at the start of the study.
  3. The details of the statistical test performed on the data in Tables 2 and 3 should be reported.

Author Response

Response to Reviewer #2’s Comments

  1. The authors should also include other studies in the discussion that describe CTCs and TWIST+ CTCs as prognostic biomarkers. Moreover, the authors should mention at the start and in relevant places the uniqueness of cohort from a different geographical location as the strength of their study.
  • We already included the results of meta-analysis about the association between CTCs and overall survival in patients with gastric cancer in the Discussion section. Because there have been few studies on the association between gastric cancer and TWIST (+) CTCs, we have added the results of previous studies that patients with TWIST (+) CTCs had lower overall survival in breast cancer and hepatocellular carcinoma as follows:

In the present study, although the difference was not statistically significant, TWIST (+) CTCs tended to be associated with shorter survival. This result is similar to the results of previous studies that patients with TWIST overexpression in CTCs had lower overall survival in breast cancer and hepatocellular carcinoma.[27,28]

  • We don’t think that geographic location is the strength of our study. Instead, we would like to stress that our study is the first study to detect TWIST (+) CTCs using FAST and to evaluate their correlation with clinicopathologic characteristics and overall survival in patients with metastatic GC. The characteristics of FAST, a size-base isolation kit, are already described in the Discussion section. Therefore, we added the following sentence in the first paragraph of the Discussion section.

In the present study, we evaluated CTCs and TWIST expression in CTCs using FAST in 31 patients with metastatic GC; CTCs and TWIST (+) CTCs were detected in 25 (80.6%) and 24 (77.4%) patients, respectively. Patients with CTCs >7.5/7.5 mL of blood showed shorter overall survival than those with CTCs ≤7.5/7.5 mL of blood. In addition, patients with TWIST (+) CTCs >2.5/7.5 mL of blood tended to show shorter overall survival than those with TWIST (+) CTCs ≤2.5/7.5 mL of blood. To the best of our knowledge, this is the first study to detect TWIST (+) CTCs using FAST and to evaluate their correlation with clinicopathologic characteristics and overall survival in patients with metastatic GC.

  1. The authors should also report if the 31 enrolled patients have any underlying disease or co-morbidity at the start of the study.
  • We have added the underlying or comorbid diseases in the Results section as follows.

Twelve patients had hypertension, 2 did diabetes mellitus, 2 did chronic hepatitis B, 2 did a previous history of pulmonary tuberculosis, 1 did Parkinson’s disease, and 1 did benign prostate hypertrophy.

  1. The details of the statistical test performed on the data in Tables 2 and 3 should be reported.
  • We have added the statistical tests used into the footnotes of Tables 2 and 3.

In Table 2, *p values are from Wilcoxon rank-sum test.

In Table 3, *p values are from Fisher exact test or x2 test.

This manuscript is a resubmission of an earlier submission. The following is a list of the peer review reports and author responses from that submission.

Round 1

Reviewer 1 Report

In this study, the authors analyzed by their original previously developed and described FAST approach, the detection of CTCs in blood samples from patients with metastatic gastric cancer. In addition, thanks to this methodology they were able to detect CTC with positive TWIST immunostaining.

The study is well performed concerning the technical aspects and the possibility to characterize in an extensive manner CTCs on solid support is an advantage of the methodology.

Nevertheless, the biological question is not innovative and not well addressed; this is mainly due to the selection of a non-optimal cohort.  Indeed, the analysis of CTCs in this specific population has already been widely described and literature data demonstrate a relation between high CTC content and poor prognosis. In addition, as specified by the authors, the cohort size is too small (31 patients, one sampling time only) and too heterogeneous (operated and non-operated patients, 5 patients with no chemotherapy) to draw conclusions. An increase in the number of patients with homogeneous clinical characteristic is required; this number of subjects needed may be statistically calculated based on the results obtained during the presented study.

In addition, a more detailed discussion concerning the results of figure 3 is mandatory:

- What are the characteristics of patients without CTC?

- What are the characteristics of patients with different counts of total CTC and TWIST(+) CTC?

Finally to properly interpret the prognosis role of CTCs and TWIST (+) CTCs on metastatic gastric cancer, correlation between clinicopathological characteristics detailed in table 1 and OS are required and a multivariate variate analysis should be made.

Author Response

Response to Reviewer #1’s Comments

  1. Nevertheless, the biological question is not innovative and not well addressed; this is mainly due to the selection of a non-optimal cohort. Indeed, the analysis of CTCs in this specific population has already been widely described and literature data demonstrate a relation between high CTC content and poor prognosis. In addition, as specified by the authors, the cohort size is too small (31 patients, one sampling time only) and too heterogeneous (operated and non-operated patients, 5 patients with no chemotherapy) to draw conclusions. An increase in the number of patients with homogeneous clinical characteristic is required; this number of subjects needed may be statistically calculated based on the results obtained during the presented study.
  • We agree with the Reviewer 1’s comment. Recently, the national gastric cancer screening program is provided by the government in persons with ≥ 40 years old every two years in Korea. Therefore, the proportion of patients with advanced gastric cancer has been markedly reduced. In addition, adjuvant chemotherapy after surgical resection is commonly performed in patients with state II or III. Therefore, the number of metastatic gastric cancer is not so common in Korea. As a result, the eligible cohort size of our study is relatively small. For including more patients in our hospital, for example, 50-60 patients, it will take about 3-4 years or more and then additional 2-3 years for survival analysis.
  • Although we already recognized the above limitation and several reports on CTCs in patients with gastric cancer, the main aim of our study was to evaluate the TWIST expression in CTCs and their correlation with prognosis. To our knowledge, there have been few reports on TWIST expression in CTCs in patients with gastric cancer. Our study was a preliminary study about clinical meaning of TWIST expression in CTCs in patients with gastric cancer. We have been continuously including more patients with gastric cancer for further analysis and studies. We sincerely request the Reviewer 1’s consideration. Therefore, we changed the title as follows:

Circulating Tumor Cells and TWIST Expression in Patients with Metastatic Gastric Cancer: A Preliminary Study

  1. In addition, a more detailed discussion concerning the results of figure 3 is mandatory:

- What are the characteristics of patients without CTC?

- What are the characteristics of patients with different counts of total CTC and TWIST(+) CTC?

  • As already stated in the Result section (including Table 2 and 3), there was no differences in the CTC and TWIST (+) CTC counts according to age, sex, histopathologic type, peritoneal dissemination, hematogenous metastasis, serum tumor makers, or response to first-line chemotherapy. Therefore, there was no remarkable differences in the clinicopathologic characteristics according to the presence/absence of total CTCs and TWIST (+) CTCs and the counts of total CTCs and TWIST (+) CTCs.
  • The following graphs show no difference in overall survival of patients according to the presence/absence of total CTCs and TWIST (+) CTCs.
  •  
  1. Overall survival curves of 6 patients without circulating tumor cells (CTCs) and 25 patients with CTCs
  1. Overall survival curves of 7 patients without TWIST (+) circulating tumor cells (CTCs) and 2 patients with TWIST (+) CTCs
  1. Finally to properly interpret the prognosis role of CTCs and TWIST (+) CTCs on metastatic gastric cancer, correlation between clinicopathological characteristics detailed in table 1 and OS are required and a multivariate variate analysis should be made.

As the Reviewer 1 indicated, we have performed the multivariate analyses of clinicopathologic factors for overall survival and then have added the following section and Table 4.

3.5. Analysis of Other Prognostic Factors

Uni- and multivariate analyses were performed to evaluate the effect of clinicopathologic factors on overall survival. Older age (>65 years), de novo status, and elevated CA 19-9 (>37 U/ml) were correlated with shorter overall survival. Multivariate analyses showed that peritoneal dissemination, elevated CA 19-9 (>37 U/ml), and non-administration of chemotherapy were independent prognostic factors (Table 4).

Table 4. Univariate and multivariate analyses of clinicopathologic factors for overall survival in the 31 patients with metastatic gastric cancer

Clinicopathologic factors

Univariate analysis

Multivariate analysis

p value

HR (95% CI)

p value

Sex (M vs. F)

0.956

0.860 (0.279−2.651)

0.793

Age (≤65 years vs. >65 years)

0.033

0.428 (0.127−1.436)

0.169

Disease status (de novo vs. recurrence)

0.008

2.458 (0.561−10.777)

0.233

Histopathologic type (intestinal vs. diffuse)

0.051

2.756 (0.976−7.781)

0.056

Peritoneal dissemination

0.331

4.293 (1.155−15.962)

0.030

Hematogenous metastasis

0.183

1.281 (0.440−3.732)

0.650

Serum CEA (≤5 ng/mL vs. >5 ng/mL)

0.711

0.997 (0.328−3.029)

0.996

Serum CA19-9 (≤37 U/mL vs. >37 U/mL)

0.005

7.531 (2.301−24.645)

0.001

Chemotherapy

0.144

0.154 (0.028−0.853)

0.032

CTC count/7.5 ml of whole blood (≤7.5 vs. >7.5)

0.054

0.848 (0.222−3.245)

0.810

TWIST (+) CTC count/7.5 ml of whole blood (≤2.5 vs. >2.5)

0.111

0.723 (0.167−3.127)

0.664

HR, hazard ratio; CI, confidence interval

Reviewer 2 Report

In this work, the authors have evaluated the presence of circulating tumor cells and TWIST+ CTCs in the peripheral blood of gastric cancer patients. Further, a strong correlation was noted between the levels of CTCs and TWIST+ CTCs with the overall survival of gastric cancer patients. Considering the small sample size used in the study, the findings are striking. The study is designed well and the results highlight the potential of CTCs as prognostic markers. 

I have two minor comments on the manuscript:

1. The number of subjects in each group in the survival curves should be indicated either in the figure legend or figure itself.

2. Were there any survival differences between the samples having no detectable CTCs with subjects having different level of CTCs?

Author Response

Dear Editor Andrada-Sorana Tripon,

On behalf of my co-authors, I am grateful for the opportunity to submit a revised version of our manuscript, with the title “Circulating Tumor Cells and TWIST Expression in Patients with Metastatic Gastric Cancer” for consideration for publication in Journal of Clinical Medicine.

Please find our point-by-point responses to the comments of the reviewers. We are thankful for the reviewers’ insightful comments.

Please let me know if we can provide any further information. We believe that our manuscript is now suitable for publication and look forward to your decision.

Thank you again for your time and consideration.

Yours Sincerely

Gwang Ha Kim

Department of Internal Medicine, Pusan National University School of Medicine and Biomedical Research Institute, Pusan National University Hospital, 179, Gudeok-ro, Seo-Gu, Busan 602-739, Korea

E-mail: doc0224@pusan.ac.kr

Response to Reviewer #2’s Comments

  1. The number of subjects in each group in the survival curves should be indicated either in the figure legend or figure itself.
  • According to the Reviewer #2’s comments, we added the number of subjects in each group in figure 4 legend.

Figure 4. (A) Overall survival curves of 18 patients with ≤ 7.5 circulating tumor cells (CTCs) and 13 patients with > 7.5 CTCs/7.5 mL of whole blood. (B) Overall survival curves of 18 patients with ≤ 2.5 TWIST (+) CTCs and 13 patients with >2.5 TWIST (+) CTCs/7.5 mL of whole blood.

  1. Were there any survival differences between the samples having no detectable CTCs with subjects having different level of CTCs?
  • There were no differences in overall survival rates of patients according to the presence/absence of CTCs and TWIST (+) CTCs. The overall survival curves are as follows:
  •  
  1. Overall survival curves of 6 patients without circulating tumor cells (CTCs) and 25 patients with CTCs
  1. Overall survival curves of 7 patients without TWIST (+) circulating tumor cells (CTCs) and 2 patients with TWIST (+) CTCs

Round 2

Reviewer 1 Report

In their revised manuscript the authors performed univariate and multivariate analyses. These studies clearly show the heterogeneity of the population. Indeed,  chemotherapy is directly associated with patient outcome. 

I fully understand the difficulties related to patient recruitment, however, due to its small size and heterogeneity, and despite the potential of the developed technique , the selected cohort does not allow the evaluation of TWIST impact in patients with metastatic gastric cancer.